# Effectiveness of Partially Hydrolyzed Guar Gum on Cognitive Function and Sleep Efficiency in Healthy Elderly Subjects in a Randomized, Double-Blind, Placebo-Controlled, and Parallel-Group Study

**DOI:** 10.3390/nu16081211

**Published:** 2024-04-19

**Authors:** Aya Abe, Mahendra Parkash Kapoor, So Morishima, Makoto Ozeki, Norio Sato, Tsuyoshi Takara, Yuji Naito

**Affiliations:** 1Department of Research and Development, Nutrition Division, Taiyo Kagaku Co., Ltd., 1-3 Takaramachi, Yokkaichi, Mie 510-0844, Japan; mkapoor@taiyokagaku.co.jp (M.P.K.); smorishima@taiyokagaku.co.jp (S.M.); mozeki@taiyokagaku.co.jp (M.O.); nsato@taiyokagaku.co.jp (N.S.); 2Medical Corporation Seishinkai, Takara Clinic, 9F Taisei Bldg., 2-3-2, Higashi-gotanda, Shinagawa, Tokyo 141-0022, Japan; t-takara@takara-clinic.com; 3Department of Human Immunology and Nutrition Science, Kyoto Prefectural University of Medicine, Kamigyoku, Kyoto 602-8566, Japan; ynaito@koto.kpu-m.ac.jp

**Keywords:** partially hydrolyzed guar gum, cognitive functions, visual memory, sleep efficiency, frailty

## Abstract

The consumption of functional foods in a daily diet is a promising approach for the maintenance of cognitive health. The present study examines the effects of water-soluble prebiotic dietary-fiber, partially hydrolyzed guar gum (PHGG), on cognitive function and mental health in healthy elderly individuals. Participants consumed either 5 g/day of PHGG or a placebo daily for 12 weeks in this randomized, double-blind, placebo-controlled, and parallel-group study. An assessment of cognitive functions, sleep quality, and subjective mood evaluations was performed at baseline and after 8 and 12 weeks of either PHGG or placebo intake. The visual memory scores in cognitive function tests and sleepiness on rising scores related to sleep quality were significantly improved in the PHGG group compared to the placebo group. No significant differences were observed in mood parameters between the groups. Vigor–activity scores were significantly improved, while the scores for Confusion–Bewilderment decreased significantly in the PHGG group when compared to the baseline. In summary, supplementation with PHGG was effective in improving cognitive functions, particularly visual memory, as well as enhancing sleep quality and vitality in healthy elderly individuals (UMIN000049070).

## 1. Introduction

Japan is one of the countries where aging is progressing most rapidly in the world. According to a report by the Ministry of Internal Affairs and Communications Statistics Bureau in 2023, the elderly population stands at 29.1% (e.g., 36.23 million) [1]. The population and proportion of the elderly are also increasing globally and are predicted to reach more than 1.6 billion by 2050 [2]. Alongside this demographic trend, cognitive decline, primarily attributed to aging, has emerged as a significant societal issue. The decline in cognitive function significantly deteriorates the quality of life by impeding independent living, increasing susceptibility to caregiving needs, and heightening the risk of progressing toward physical frailty [3]. As cognitive function declines, it progresses to dementia. It is estimated that by 2025, approximately 20% of the elderly population, or one in every five individuals, will develop dementia [4]. Similarly, this trend is not limited to Japan; dementia cases are rapidly rising worldwide. According to the World Health Organization (WHO), there are over 55 million dementia patients worldwide, with approximately 10 million new cases reported annually [5]. As of 2019, the economic cost of dementia amounted to USD 1.3 trillion annually [5]. Dementia not only affects the individuals themselves but also has physical, psychological, social, and economic impacts on caregivers, families, and society as a whole. Reducing the risk of dementia is a global challenge that requires urgent attention and action.

While the detailed mechanisms underlying the onset of cognitive decline remain unclear, oxidative stress, inflammation, alterations in neurotransmitters, and phosphorylation of β-amyloid and tau proteins in the brain have been entailed [6,7]. Obesity and lifestyle-related diseases are mostly considered to raise the risk of developing dementia [8,9]. Correcting unhealthy lifestyle habits is undoubtedly important, wherein actively consuming functional food components that have a positive impact on brain function would be beneficial [10,11]. Among these, with the advancing comprehension of the bidirectional relationship between the brain and the gut [12,13,14], probiotics and prebiotics, recognized for their capacity to enhance gut health, are considered as potential candidates. Intestinal bacteria produce various metabolites such as short-chain fatty acids (SCFAs), secondary bile acids, amino acid metabolites, and neurotransmitters, which can influence brain function through their effects on the nervous and immune systems [15,16]. Thus, improving the intestinal environment is anticipated to contribute to the prevention of cognitive decline. Several human clinical trials focusing on the brain–gut connection have been conducted, revealing that consuming probiotics and prebiotics has been corroborated to yield beneficial outcomes on cognitive function [17,18,19,20,21], depression [18,22], sleep quality [23,24], mood [25], etc.

Partially hydrolyzed guar gum (PHGG), derived from guar beans grown in arid regions such as India and Pakistan, is a water-soluble dietary fiber. It is produced by hydrolyzing the high-viscosity gum extracted from the endosperm of the seeds, making it easily added to food products as a supplement [26]. Structurally composed of galactomannan, PHGG exhibits high fermentability by intestinal bacteria, which promotes the production of short-chain fatty acids [27,28] and contributes to maintaining the intestinal barrier function [29,30]. Numerous human clinical trials have demonstrated its efficacy in improving bowel movements [31,32] and alleviating diarrhea [33,34,35], leading to its widespread use as a prebiotic. In a previous clinical study targeting healthy adults, we revealed that PHGG improves mental health, including sleep and motivation, through its effect on gut microbiota [36]. Therefore, in this study, we hypothesized that PHGG also contributes to the cognitive function and mental health of the elderly and conducted a randomized controlled trial targeting healthy elderly individuals to verify this hypothesis.

## 2. Materials and Methods

### 2.1. Study Design

This study was designed as a randomized double-blind, placebo-controlled, and parallel-group trial with participants allocated in a 1:1 ratio. Taiyo Kagaku Co., Ltd. (TKC, Yokkaichi, Japan) prepared the study protocol and provided the test food. The study was conducted at Medical Corporation Seishinkai, Takara Clinic (Tokyo, Japan) and Minami-machi Clinic (Tokyo, Japan) between 12 December 2022 and 25 March 2023. This study was conducted in accordance with the ethical principles based on the Helsinki Declaration and the Ethical Guidelines for Human Medical Research. A fair review was conducted from the perspective of protection of human rights and safety assurance, and the study protocol was approved by the institutional review board of the Takara Clinic (approval number: 2209-00191-0022-33-TC; Date of approval: 14 September 2023) and registered at the UMIN-CTR (Trial ID: UMIN000049004). The protocol was not modified from the time of the final setup or during the study.

### 2.2. Study Functional Food Material

The commercially available PHGG (Sunfiber^®^) used in this study was supplied by TKC. The daily dosage of PHGG was set at 5 g, with a water-soluble dietary-fiber content exceeding 80%, as determined by the Association of Official Agricultural Chemists (AOAC) method for measuring the soluble-fiber content. This dosage had been previously validated in human trials for its efficacy in enhancing gut health and promoting mental well-being, including improvements in aspects such as sleep and motivation [36]. The placebo consisted of maltodextrin.

### 2.3. Study Participants

Inclusion criteria for the participants consisted of the following: (1) healthy Japanese individuals of both genders aged over 60 years; (2) subjects with a Japanese version of the Mini Mental State Examination (MMSE-J) [37] score of 24 or higher at screening; and (3) individuals exhibiting a relatively low gait speed. Exclusion criteria included: (1) subjects undergoing medical treatment or with a medical history of malignant tumors, heart failure, and myocardial infarction; (2) participants with a pacemaker or implantable cardioverter defibrillator (ICD); (3) individuals currently undergoing treatment for chronic diseases such as cardiac arrhythmia, liver disorder, kidney disorder, cerebrovascular disorder, rheumatism, diabetes mellitus, dyslipidemia, hypertension, or any other chronic conditions; (4) subjects habitually consuming health-promoting foods; (5) participants currently taking medications (including herbal medicines) and supplements; (6) individuals consciously taking dietary fiber and oligosaccharides; (7) subjects allergic to medications and/or test food-related products; (8) participants with an exercise habit of two or more times per week for at least 30 min each time, consistently for one year or longer; (9) subjects suffering from COVID-19; (10) participants enrolled in other clinical trials within the last 28 days before agreeing to participate in this trial or planning to participate in another trial during this trial; (11) individuals deemed ineligible to participate in this study by the physician. The recruitment of study participants was conducted between 15 September 2022 and 19 November 2022 through the monitor recruitment website operated by ORTHOMEDICO, called Go Toroku (https://www.go106.jp/, accessed on 1 September 2022).

### 2.4. Intervention and Outcomes

Since there were no prior human trials on the test food’s effects on cognition and limited data on other prebiotics, we modeled our trial schedule on similar probiotic studies, assessing at 8 [38] and 12 weeks [17]. Accordingly, our intake period was set to 12 weeks, with an interim evaluation at 8 weeks. The participants took either PHGG or a placebo dissolved in water every day with breakfast for 12 consecutive weeks. The primary outcome was the evaluation of cognitive function using the Cognitrax test [39]. Additionally, physical function, quality of life (QOL) assessed through a questionnaire on sleep, stress, and fatigue, and the safety of the study food were assessed. At the pre-study examination, as well as after 8 and 12 weeks of consuming their respective treatments, the participants visited the clinical center and underwent examinations. Throughout the study period, participants were instructed to follow the prescribed consumption of the test food, avoid making significant lifestyle changes, refrain from alcohol and excessive exercise before examinations, fast for at least 6 h before blood sampling (with only water consumption allowed), promptly report any health changes, and minimize intake of specific health foods and dietary components.

### 2.5. Cognitrax Test

Cognitive function was assessed using the Cognitrax test [39], which is a computer-based neurocognitive assessment test including a verbal memory test. It is characterized by its sensitivity in milliseconds, accuracy, and high reliability, as well as having low learning and ceiling effects. Through the evaluation of the following ten types of wide-ranging functional areas, it is possible to thoroughly investigate which areas are experiencing a decline in function. (1) The verbal memory test evaluates the memory function for words. (2) The visual memory test evaluates the memory function for shapes. (3) The finger tapping test assesses motor speed by evaluating how quickly a key can be tapped. (4) The symbol digit coding test evaluates cognitive flexibility, attention, and information processing speed by referencing a given symbol-to-number chart and inputting the corresponding number as quickly as possible. (5) The Stroop test evaluates executive functions, simple and complex reaction speeds, and information processing speed through three tests: pressing a key when a word appears, pressing a key when the meaning of the words and their colors match, and pressing a key when the words and colors do not match. (6) The shifting attention test evaluates executive functions, reaction speed, and information processing speed by measuring the ability to quickly and accurately switch from one instruction to another. (7) The continuous performance test measures sustained attention over a long period and evaluates sustained attention, selective reaction speed, and impulsivity. (8) The perception of emotions test assesses social cognitive ability, emotional judgment ability, and selective reaction time by measuring the ability to recognize and judge human facial expressions. (9) The non-verbal reasoning test evaluates the ability to understand theoretical construction, recognition of theories, and cognitive speed by measuring the ability to recognize relationships in visual or abstract concepts. (10) The 4-part continuous performance test evaluates working memory and sustained attention by measuring simple reaction speed, the ability to sustain attention, memory for the image one back, and memory for the image two back. The scores of each test were converted through a comparison with age-matched norms, and the standardized scores were calculated for specific cognitive domains, neurocognitive index (NCI), composite memory, verbal memory, visual memory, psychomotor speed, reaction time, complex attention, cognitive flexibility, processing speed, executive function, social acuity, reasoning, working memory, sustained attention, simple attention, and motor speed [40].

### 2.6. QOL Questionnaires

Quality of life (QOL) questionnaires assessing sleep quality and stress were evaluated using the Oguri–Shirakawa–Azumi Sleep Inventory, Middle-Aged version (OSA-MA) [41], and the Profile of Mood States 2nd Edition (POMS-2) [42]. OSA-MA consists of a four-step questionnaire about sleep state, comprising 16 questions that are categorized into the following subdomains: “sleepiness on rising”, “initiation and maintenance of sleep”, “frequent dreaming”, “refreshing”, and “sleep length”. A higher OSA-MA subscale score indicates a better sleep state [41]. POMS-2 comprises 65 questions describing seven distinct moods: “anger-hostility”, “confusion-bewilderment”, “depression-dejection”, “fatigue-inertia”, “tension-anxiety”, “vigor-activity”, and “friendliness”. Higher scores on the “vigor–activity” and “friendliness” items indicate better states, while lower scores on other items suggest better states [42].

### 2.7. Sample Size

In the absence of prior research investigating the effects of the test food on cognitive function, we determined the sample size by referencing studies related to improvements in gut health and mental well-being. In general, these studies required around 20 to 40 participants for each group [18]. Accordingly, we settled on a sample size of 30 for each group in this study, and to account for potential dropouts, we cautiously increased the sample size to 33 for each group.

### 2.8. Selection, Randomization, and Blinding

All study procedures were executed by ORTHOMEDICO Inc. (Tokyo, Japan) based on the study plan devised by TKC. The participants underwent preliminary assessments, including physical examinations, lifestyle habit surveys, cognitive function evaluations (using MMSE and Cognitrax), and a quality of life questionnaire survey (subjective assessments related to sleep and mood). In total, 66 eligible participants were selected from a total of 91 candidates based on the pre-specified inclusion and exclusion criteria. Participants were randomly assigned to either the PHGG group or the placebo group. This randomization was stratified by gender, and the allocation process was carried out by an independent researcher from ORTHOMEDICO, who had no involvement in the study’s planning, execution, or data analysis. Importantly, this allocation process was conducted in a double-blind manner, ensuring that both participants and investigators remained unaware of group assignments until the intervention was completed and data analysis had commenced.

### 2.9. Statistical Analysis

All statistical analyses in this study were two-sided, and the significance level was set at 5%. Data analyses were performed using Windows SPSS, v.23.0 (IBM Japan, Ltd., Tokyo, Japan). The mean values of the measurements for study outcomes were calculated and analyzed at each time point. Data are expressed as means ± standard deviation. The subjects’ gender for each group was compared using Fisher’s exact test, and the other items were compared using Student’s *t*-tests. For the efficacy endpoints, group comparisons of the baseline were analyzed using the Student’s *t*-test, while group comparisons of measurements during the intake period were conducted using a linear mixed-effects model. This model included baseline values as covariates, along with time points, groups, interactions between time points and groups, interactions between baseline values and time points, and participants as factors. Within-group comparisons were analyzed using a paired *t*-test for comparisons between each observation period before and after intake.

## 3. Results

### 3.1. Participants’ Characteristics

The study flow diagram is displayed in Figure 1. Ninety one healthy men and female aged over sixty were screened for this study. Sixty-six subjects were selected based on the inclusion and exclusion criteria and randomly allocated to the placebo or PHGG groups. During the allocation process, one participant in the test food group did not receive the assigned intervention. Including this participant, there were a total of five participants (three in PHGG group and two in the placebo group) who were lost to follow-up after not attending the clinic post the 8-week intake period.

For both the Full Analysis Set (FAS) and the Safety Analysis Set (SAF), four participants (two from each group) who had never received the intervention post-allocation and one participant from the test food group who did not receive the assigned intervention were excluded. Consequently, the number of participants in each analysis dataset was 61 (30 in the test food group and 31 in the placebo group). Table 1 details the demographic and baseline characteristics of the trial participants, and there were no significant differences between groups for each parameter.

### 3.2. Cognitrax

Table 2 displays cognitive domain scores calculated from the Cognitrax test in this study. A between-group comparison revealed that at 12 weeks, visual memory scores, and at 8 weeks, simple attention scores in the PHGG group, were significantly superior to those observed in the placebo group. In detail, at baseline, the visual memory scores did not significantly differ between the PHGG and placebo groups (PHGG: 96.5 ± 13.7; placebo: 93.1 ± 17.9, *p* = 0.417). However, after 12 weeks of supplementation, the PHGG group exhibited significantly higher scores compared to the placebo group (PHGG: 99.6 ± 15.8; placebo: 90.5 ± 15.4, *p* = 0.023) with a significantly larger change from baseline (PHGG: 3.1 ± 16.7; placebo: −2.7 ± 17.3, *p* = 0.023). Regarding simple memory, the scores of the PHGG group were significantly higher compared to those of the placebo group after 8 weeks supplementation (PHGG: 99.7 ± 16.5; placebo: 14.6 ± 230.2, *p* = 0.020), with a significantly larger change from baseline (PHGG: 14.8 ± 97.9; placebo: −23.4 ± 269.0, *p* = 0.020). However, despite the absence of a significant difference in baseline scores between the groups, the initial scores of the PHGG group were more than double those of the placebo group (PHGG: 84.8 ± 100.9; placebo: 38.0 ± 170.2, *p* = 0.196), indicating a significant initial variance. Upon intra-group analysis, notable enhancements were observed in both the placebo and PHGG groups across various evaluative domains, including the neurocognitive index (NCI), composite memory, verbal memory, psychomotor speed, and executive function, relative to their baseline measurements at both 8 and 12 weeks subsequent to supplementation.

### 3.3. OSA-MA Sleep Questionnaire

Table 3 presents the outcomes of the subjective evaluation of sleep assessed using the OSA-MA questionnaire. In the between-group comparison, the score of the PHGG group for sleepiness on rising after 8 weeks of supplementation was significantly superior (higher) compared to the placebo (PHGG: 20.1 ± 4.4; placebo: 19.6 ± 6.0, *p* = 0.043) with a significantly larger change from baseline (PHGG: 1.8 ± 3.6; placebo: −0.6 ± 3.9, *p* = 0.043). Additionally, the actual values and the amount of change after 12 weeks showed a tendency to be higher in the PHGG group compared to the placebo group (both *p* = 0.096). Also, in the intra-group comparison, in the PHGG group, the scores for sleepiness on rising and initiation and maintenance of sleep significantly increased after 8 weeks and 12 weeks of intake compared to baseline, whereas in the placebo group, no significant changes were observed. Furthermore, although not statistically significant, the changes from baseline after 12 weeks of intake in the items of initiation and maintenance of sleep, frequent dreaming, refreshing, and sleep length were greater in the PHGG group, consistently indicating an improvement due to PHGG intake.

### 3.4. POMS-2 Questionnaire

Table 4 shows results of the subjective moods assessment evaluated with the POMS-2 questionnaire. There were no significant differences between the groups in all the evaluated items. However, in the within-group comparisons of the PHGG group, the scores of Vigor–activity after 8 weeks of supplementation were significantly higher (baseline: 54.7 ± 10.7; week 8: 57.1 ± 9.8, *p* = 0.049), and the scores of Confusion–Bewilderment after 12w of supplementation were significantly lower than those of baseline (baseline: 47.4 ± 8.3 week 12: 45.2 ± 7.7, *p* = 0.018) indicating an improvement. In the within-group comparison of the placebo group, no significant improvements were observed in any of the measures.

### 3.5. Safety

Throughout the trial period, there were no adverse events attributed to the intake of PHGG, confirming its safety.

## 4. Discussion

In previous studies, we reported that PHGG, a highly fermentable prebiotic dietary fiber, improved gut microbiota and bowel movements in healthy adult men and women, leading to improvements in sleep quality and motivation [36]. In this study, we expanded our investigation to explore how PHGG can help prevent frailty by improving cognitive function as well as sleep quality, and mental health in elderly individuals with normal cognitive abilities, indicated by an MMSE score of 24 or higher. Cognitive function assessments and subjective evaluations of sleep and mood were conducted at baseline, 8 weeks, and 12 weeks.

As a result, we newly revealed the effectiveness of PHGG on visual memory in the cognitive function test using Cognitrax, which was set as the primary outcome. The standardized scores of Cognitrax are scored based on age-matched normative data, normalized with a mean value of 100 and a standard deviation of 15. Higher scores indicate a better performance, with scores above 109 considered “above average”, 90–109 as “average”, 80–89 as “low average”, 70–79 as “below average”, and below 70 as “low” [40]. Although the average values for both groups remained within the “average” range before and after the intervention, only in the PHGG group did the intervention bring the scores closer to the average for their age group, suggesting that the intake of PHGG improves visual memory.

The visual memory assessment in Cognitrax is conducted through a test where initially, 15 types of figures are presented on the screen, followed by a task to identify the same figures from a new set of 15 types displayed subsequently. Participants are evaluated on how accurately and swiftly they can memorize and re-recognize visual information. This capability is associated with the ability to remember seen information and recall it later, which is deeply involved in many aspects of daily life, such as remembering graphic instructions, navigating, operating machines, recalling images, and/or remember a calendar of events [40]. These abilities are considered critical functions for individuals to live independently in their daily lives. The improvement of this function through the intake of PHGG is expected to contribute to the prevention of cognitive frailty in the elderly.

In this trial, the effectiveness of PHGG was also demonstrated for the quality of sleep assessed using the OSA-MA questionnaire. The OSA-MA questionnaire is a psychological scale that evaluates the intra sleep phase upon waking and is applicable to elderly individuals as well, with higher values indicating a better state [41]. In the PHGG group, when compared to baseline, there was a significant improvement in scores for sleepiness on rising and initiation and maintenance of sleep, with sleepiness on rising also showing a significant improvement compared to the placebo group. In the OSA-MA questionnaire, sleepiness on rising is a composite measure that combines the scores for concentration, stress relief, mental clarity, and effortlessness [41]. Although no significant intergroup differences were observed in the changes from baseline for these individual scores, the PHGG group consistently showed higher values across all items compared to the placebo group, indicating a uniform direction towards improvement. Above all, it can be inferred that the consumption of PHGG resulted in an improvement in sleepiness on rising characterized by enhanced mental clarity, reduced stress levels, decreased irritability, and improved ability to concentrate. Furthermore, these results are consistent with those of a clinical trial involving healthy subjects, where the quality of sleep was assessed using a visual analog scale, and the intake of PHGG significantly improved the scores for refreshment on waking up and fatigue on waking up [36]. Given that an improvement in sleep quality was observed despite differences in participant backgrounds and assessment methods, it suggests that the intake of PHGG is likely to enhance the quality of sleep.

Additionally, subjective evaluations of mood conducted with POMS-2 demonstrated that the intake of PHGG positively influenced mood states. Although no significant differences were observed between groups, in the PHGG group, the values for vigor and vitality were significantly higher at 8 weeks post-intake compared to baseline, while the values for confusion and bewilderment were significantly lower. Previous clinical trials in healthy subjects have also shown that PHGG intake may enhance motivation toward work and study, and our results support these findings. While no significant differences were found in the individual items of POMS-2, it is considered that this may be due to large individual differences in the scores of each assessment item. It is anticipated that future studies, by refining participant conditions and survey evaluation methods, can more accurately assess the effectiveness of PHGG intake on mood. In summarizing these collective findings, consistent with previous findings suggesting that the intake of PHGG contributes to mental health through the improvement of gut microbiota, this current study also demonstrates that PHGG intake improves the quality of sleep and vitality in healthy elderly individuals. Furthermore, this trial has newly presented the possibility that PHGG intake may have positive effects on cognitive function.

PHGG is a fermentable dietary fiber that has been shown to improve bowel movements [31,32] and alleviates diarrhea [33,34,35], increase beneficial bacteria such as Bifidobacterium and butyrate-producing bacteria [34,35,43,44], and promote the production of short-chain fatty acids [45]. Clinical trials targeting middle-aged to elderly individuals have also confirmed the prebiotic effects of PHGG. In a randomized single-blinded, placebo-controlled trial involving long-term care facility residents (*n* = 52) aged 83.9 ± 7.6 years, a daily intake of 5 g/day of PHGG resulted in significantly less laxative use than placebo [46]. Similarly, in a clinical trial involving 39 chronic constipation patients aged 56.26 ± 16.21 years, daily intake of 5 g/day of PHGG for 4 weeks resulted in increased spontaneous bowel movements, improved stool consistency, shortened colonic transit time, reduced frequency of abdominal pain, and decreased use of laxatives [47]. Additionally, in a double-blinded, placebo-controlled trial involving 100 postoperative patients aged 59 to 70 years using enteral nutrition, intake of 20 g/day of PHGG for 5 or more days led to a reduction in the incidence of diarrhea and a decrease in the frequency of discontinuation of enteral nutrition due to diarrhea [48]. Furthermore, in a clinical trial involving 15 dialysis patients aged 63.9 ± 1.8 years, daily intake of 12 g/day of PHGG for 4 weeks resulted in improved stool consistency, increased levels of fecal beneficial bacteria such as Bifidobacterium and Clostridium cluster XVIII, and increased levels of fecal short-chain fatty acids [49].

Short-chain fatty acids are recognized as crucial factors in the gut–brain axis, suggesting that the improvements in sleep quality, vigor, and cognitive function associated with PHGG intake may be influenced by the promotion of intestinal SCFA production by PHGG. SCFA confer neuroprotective effects through mechanisms such as strengthening the intestinal barrier function, suppressing systemic chronic inflammation, enhancing blood–brain barrier integrity, regulating immune cells, and inhibiting brain inflammation [50,51,52]. Additionally, hormones secreted in the gut due to their promotion by short-chain fatty acids also modulate the central nervous system via systemic circulation or the vagus nerve pathway [53,54]. Indeed, PHGG has been shown to enhance the intestinal barrier function by promoting the production of short-chain fatty acids, thereby increasing mucin and tight junction proteins and inhibiting the influx of inflammatory substances into the body, consequently suppressing internal inflammation [29,30,55,56,57,58]. In addition, the results of fecal culture from elderly participants supplemented with PHGG revealed a change in the profile of the fecal microbiota, along with enhanced production of short-chain fatty acids in the fermentation supernatant, and the addition of this fermentation supernatant was found to strengthen the barrier function of inflammation-induced intestinal epithelial model cells [59]. Essentially, these results suggest that PHGG may alter the intestinal environment in elderly individuals, thus potentially helping to maintain intestinal barrier function and possibly reducing the risk of bacterial translocation and toxin influx into the body.

Moreover, research in a mouse model of depression induced by unpredictable stress in 6-week-old mice has shown that the intake of 600 mg/kg/day of PHGG for one month increases serotonin and dopamine levels in the serum, hippocampus, and striatum of the brain, consequently improving depressive behavior [60]. Additionally, in aging model rats induced with oxidative stress by injecting D-galactose at 14 weeks of age, supplementation with 500, 1000, and 1500 mg/kg/day of PHGG for 10 weeks enhances the activity of antioxidant enzymes, alleviates oxidative damage in the liver and hippocampus, and increases brain-derived neurotrophic factor (BDNF) and choline acetyltransferase [61]. Serotonin and dopamine signaling are neurotransmitters involved in mood, memory, learning, and sleep [62,63,64,65,66], while BDNF promotes neuronal growth and regeneration [67], and choline acetyltransferase regulates the synthesis of acetylcholine, a neurotransmitter involved in signaling at neuromuscular junctions [68]. These substances play crucial roles in fundamental brain processes such as learning, memory, consciousness, and sleep. PHGG may regulate these substances in the brain and thus modulate the brain function. Thus, PHGG has the potential to modulate the gut–brain-axis multifaceted manner through various mechanisms, potentially exerting positive effects on the brain function. However, this study has several limitations. While it was standardized and reliable, it relied solely on subjective evaluations to assess cognitive function, sleep, and mood. Moreover, the intervention period of three months was relatively short, limiting the evaluation of PHGG’s long-term effects. Additionally, the study had limitations in terms of sample size. Given that cognitive decline is a gradual process that unfolds over the long term, future research should aim for more comprehensive studies with larger sample sizes and longer durations, incorporating objective measures as well.

## 5. Conclusions

Supplementation with PHGG has shown potential effectiveness in improving cognitive function, particularly visual memory in healthy elderly individuals. This improvement is crucial for maintaining independence and quality of life, as memory plays a pivotal role in daily activities and social interactions. In addition, the study highlighted the beneficial effects of PHGG on sleep quality, suggesting that PHGG supplementation could contribute to a more active, engaged, and fulfilling lifestyle in aging individuals. However, further comprehensive studies, including larger-scale investigations and detailed mechanism substantiations, are warranted in the future.

## Figures and Tables

**Figure 1 nutrients-16-01211-f001:**
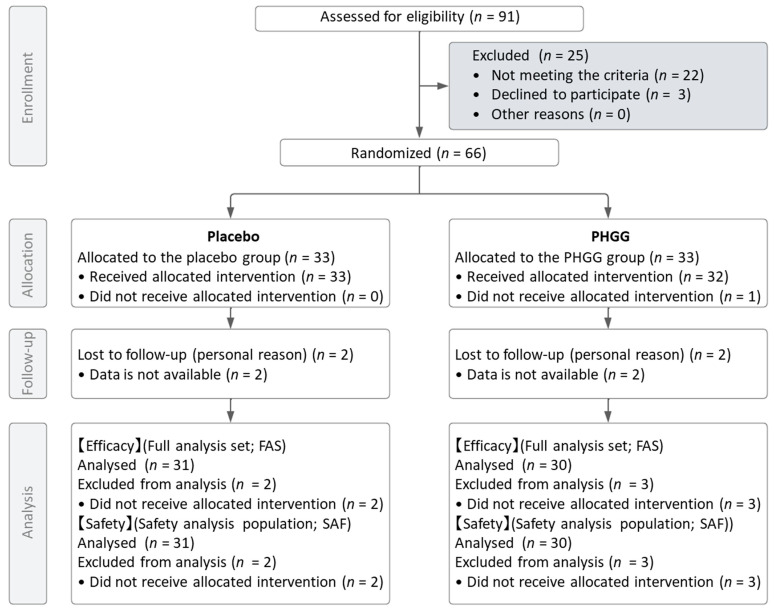
Flow chart of study subjects.

**Table 1 nutrients-16-01211-t001:** Subject demographics at baseline.

	Placebo	PHGG	*p*-Value
	(*n* = 31)	(*n* = 30)	
Gender (Male/Female)	16/15	15/15	>0.999
Age (years)	68.1 ± 7.2	66.8 ± 6.1	0.453
Height (cm)	163.2 ± 8.4	160.5 ± 9.0	0.231
Weight (kg)	62.1 ± 14.1	59.5 ± 11.7	0.452
Body fat (%)	26.1 ± 8.1	26.6 ± 7.4	0.778
Body Mass Index (kg/m^2^)	23.1 ± 4.1	22.9 ± 3.1	0.832
Systolic blood pressure (mmHg)	132.2 ± 17.3	132.8 ± 17.2	0.892
Diastolic blood pressure (mmHg)	83.1 ± 11.7	83.4 ± 11.5	0.910
MMSE score	28.7 ± 1.4	28.9 ± 1.2	0.575

Data are represented as mean ± SD. The subjects’ genders for each group were compared using Fisher’s exact test, and the other items were compared using Student’s *t*-tests. No significant differences were detected between the placebo and PHGG groups.

**Table 2 nutrients-16-01211-t002:** Results of Cognitrax test.

		Baseline	Week 8	Week 12	Change from Baseline	
					Week 8	Week 12
Neurpcognition index (NCI)	Placebo (*n* = 31)	89.5 ± 16.5	93.8 ± 17.9 ^#^	100.9 ± 11.1 ^##^	4.3 ± 12.7	11.3 ± 12.4
	PHGG (*n* = 30)	98.1 ± 16.4	103.4 ± 11.5 ^##^	105.4 ± 11.2 ^##^	5.3 ± 8.9	7.4 ± 10.0
	*p*-value	**0.048 ***	0.131	0.878	0.131	0.878
Composite memory	Placebo (*n* = 31)	87.6 ± 21.4	93.6 ± 18.3	96.5 ± 17.4 ^##^	6.0 ± 18.4	8.8 ± 20.0
	PHGG (*n* = 30)	94.3 ± 19.9	97.9 ± 17.0	102.9 ± 16.9 ^##^	3.6 ± 15.0	8.6 ± 15.2
	*p*-value	0.211	0.835	0.369	0.835	0.369
Verbal memory	Placebo (*n* = 31)	86.5 ± 23.9	95.1 ± 22.8 ^#^	103.5 ± 19.0 ^##^	8.5 ± 23.6	17.0 ± 25.4
	PHGG (*n* = 30)	94.4 ± 23.2	98.8 ± 20.7	105.8 ± 19.0 ^##^	4.5 ± 16.8	11.4 ± 16.3
	*p*-value	0.198	0.903	0.826	0.903	0.826
Visual memory	Placebo (*n* = 31)	93.1 ± 17.9	95.0 ± 13.4	90.5 ± 15.4	1.9 ± 16.3	−2.7 ± 17.3
	PHGG (*n* = 30)	96.5 ± 13.7	98.1 ± 13.1	99.6 ± 15.8	1.6 ± 13.9	3.1± 16.7
	*p*-value	0.417	0.597	**0.023 ***	0.597	**0.023 ***
Psychomotor speed	Placebo (*n* = 31)	87.1 ± 32.9	96.5 ±24.1 ^#^	105.1 ± 17.1 ^##^	9.4 ± 28.1	18.0 ± 27.8
	PHGG (*n* = 30)	101.4 ± 23.9	107.9 ± 16.9 ^#^	109.6 ± 15.1 ^#^	6.5 ± 21.2	8.2 ± 17.7
	*p*-value	0.057	0.159	0.970	0.159	0.970
Reaction time	Placebo (*n* = 31)	101.8 ± 28.9	95.8 ± 22.5	100.7 ± 21.4	−6.0 ± 24.8	−1.1 ± 18.4
	PHGG (*n* = 30)	96.3 ± 16.5	101.4 ± 20.6	104.5 ± 18.7 ^#^	5.1 ± 16.0	8.2 ± 20.8
	*p*-value	0.364	0.056	0.133	0.056	0.133
Complex attention	Placebo (*n* = 31)	81.5 ± 40.4	85.4 ± 52.8	100.1 ± 23.4 ^#^	3.9 ± 48.4	18.6 ± 27.6
	PHGG (*n* = 30)	99.5± 26.5	106.6 ± 13.7	105.2 ± 24.4	7.0 ± 23.3	5.7 ± 26.8
	*p*-value	**0.043 ***	0.117	0.729	0.117	0.729
Cognitive flexibility	Placebo (*n* = 31)	89.9 ± 16.8	97.9 ± 15.6 ^##^	101.9 ± 13.8 ^##^	8.0 ± 17.3	12.0 ± 16.9
	PHGG (*n* = 30)	98.6 ± 18.2	103.2 ± 14.2 ^#^	104.8 ± 14.8 ^##^	4.6 ± 11.6	6.1 ± 11.6
	*p*-value	0.057	0.769	0.654	0.769	0.654
Processing speed	Placebo (*n* = 31)	106.7 ± 14.2	107.6 ± 14.5	112.0 ± 11.9 ^##^	0.9 ± 8.7	5.3 ± 9.1
	PHGG (*n* = 30)	111.3 ± 12.3	111.9 ± 15.5	115.9 ± 14.0 ^##^	0.6 ± 8.7	4.6 ± 7.5
	*p*-value	0.185	0.986	0.874	0.986	0.874
Executive function	Placebo (*n* = 31)	91.7 ± 15.7	98.3 ± 14.2 ^#^	102.8 ± 12.3 ^##^	6.5 ± 16.2	11.1 ± 15.8
	PHGG (*n* = 30)	98.2 ± 18.3	103.0 ± 14.5 ^##^	104.4 ± 15.2 ^##^	4.9 ± 11.2	6.3 ± 11.4
	*p*-value	0.147	0.619	0.579	0.619	0.579
Social acuity	Placebo (*n* = 31)	79.3 ± 26.4	88.5 ± 19.1 ^#^	89.7 ± 22.0 ^#^	9.3 ± 28.0	10.5 ± 25.5
	PHGG (*n* = 30)	91.0 ± 23.7	92.0 ± 21.5	91.6 ± 20.3	1.0 ± 21.7	0.6 ± 22.2
	*p*-value	0.073	0.939	0.571	0.939	0.571
Reasoning	Placebo (*n* = 31)	91.6 ± 14.4	100.6 ± 12.2 ^##^	97.7 ± 15.7	9.0 ± 15.9	6.1 ± 22.1
	PHGG (*n* = 30)	99.7±15.4	98.5 ± 16.8	97.4 ± 18.9	−1.2 ± 19.3	−2.3 ± 19.0
	*p*-value	**0.038 ***	0.293	0.624	0.293	0.624
Working memory	Placebo (*n* = 31)	103.8 ± 13.4	102.8 ± 17.2	106.3 ± 15.6	−0.9 ± 18.1	2.5 ± 13.6
	PHGG (*n* = 30)	108.9 ± 11.5	110.6 ± 13.7	107.9 ± 12.7	1.8 ± 10.0	−0.9 ± 12.4
	*p*-value	0.117	0.159	0.674	0.159	0.674
Sustained attention	Placebo (*n* = 31)	100.2 ± 20.7	100.0 ± 21.8	105.8 ± 20.9	−0.2 ± 17.2	5.6 ± 18.6
	PHGG (*n* = 30)	109.9 ± 17.0	109.9 ± 15.9	110.5 ± 10.3	0.1 ± 16.4	0.6 ± 17.0
	*p*-value	0.051	0.316	0.907	0.316	0.907
Simple attention	Placebo (*n* = 31)	38.0 ± 170.2	14.6 ± 230.2	76.9 ± 88.7	−23.4 ± 269.0	38.9 ± 152.5
	PHGG (*n* = 30)	84.8 ± 100.9	99.7 ± 16.5	88.2 ± 68.6	14.8 ± 97.9	3.4 ± 122.9
	*p*-value	0.196	**0.020 ***	0.923	**0.020 ***	0.923
Motor speed	Placebo (*n* = 31)	78.6 ± 39.9	90.3 ± 26.4 ^#^	99.3 ± 17.6 ^##^	11.7 ± 37.4	20.6 ± 37.6
	PHGG (*n* = 30)	94.5 ± 27.9	103.0 ± 15.7 ^#^	102.3 ± 15.0	8.5 ± 27.5	7.7 ± 22.7
	*p*-value	0.076	0.064	0.960	0.064	0.960

Data are presented as mean ± SD. The scores for each item were calculated through a comparison with age-matched norms. Between-group comparisons of baseline were conducted using the Student’s *t*-test, while comparisons at weeks 8 and 12 were analyzed using a linear model that included baseline values as covariates, along with time points, groups, interactions between time points and groups, interactions between baseline values and time points, and participants as factors. Significant scores are presented in bold letters. Within-group comparisons were performed using paired *t*-tests. (Keys: *: significant at *p* ≤ 0.05 (between-group comparison with placebo); ^#^: significant at *p* ≤ 0.05 (within-group comparison with baseline); ^##^: significant at *p* ≤ 0.01 (within-group comparison with baseline)).

**Table 3 nutrients-16-01211-t003:** Results of OSA-MA sleep questionnaire.

		Baseline	Week 8	Week 12	Change from Baseline	
					Week 8	Week 12
Sleepiness on rising	Placebo (*n* = 30)	20.2 ± 6.7	19.6 ± 6.0	19.9 ± 6.6	−0.6 ± 3.9	−0.3 ± 4.1
	PHGG (*n* = 31)	18.4 ± 4.6	20.1 ± 4.4 ^##^	20.0 ± 4.2 ^##^	1.8 ± 3.6	1.7 ± 3.5
	*p*-value	0.204	**0.043 ***	0.096	**0.043 ***	0.096
Initiation and maintenance of sleep	Placebo (*n* = 30)	17.9 ± 6.0	17.5 ± 5.8	18.6 ± 5.4	−0.4 ± 5.0	0.7 ± 4.2
	PHGG (*n* = 31)	17.0 ± 5.3	18.6 ± 4.9 ^#^	19.2 ± 5.1 ^##^	1.6 ± 4.2	2.2 ± 5.0
	*p*-value	0.518	0.124	0.286	0.124	0.286
Frequent dreaming	Placebo (*n* = 30)	23.1 ± 7.0	21.8 ± 6.2	22.8 ± 5.9	−1.2 ± 4.3	−0.3 ± 5.5
	PHGG (*n* = 31)	22.3 ± 6.1	22.8 ± 5.4	22.7 ± 6.0	0.5 ± 5.3	0.4 ± 4.6
	*p*-value	0.644	0.194	0.761	0.194	0.761
Refreshing	Placebo (*n* = 30)	20.6 ± 6.6	21.6 ± 6.4	20.9 ± 6.6	1.0 ± 5.3	0.3 ± 5.3
	PHGG (*n* = 31)	19.7 ± 5.2	20.3 ± 4.9	20.9 ± 4.8	0.7 ± 3.7	1.2 ± 4.5
	*p*-value	0.547	0.551	0.646	0.551	0.646
Sleep length	Placebo (*n* = 30)	20.7 ± 5.9	19.0 ± 6.4	18.8 ± 6.4	−1.7 ± 5.8	−2.0 ± 6.5
	PHGG (*n* = 31)	19.8 ± 4.4	19.7 ± 4.0	20.0 ± 4.6	−0.1 ± 4.2	0.3 ± 4.5
	*p*-value	0.474	0.339	0.169	0.339	0.169

Data are represented as mean ± SD. Between-group comparisons of baseline were conducted using the Student’s *t*-test, while comparisons at weeks 8 and 12 were analyzed using a linear model that included baseline values as covariates, along with time points, groups, interactions between time points and groups, interactions between baseline values and time points, and participants as factors. Significant values are presented in bold letters. Within-group comparisons were performed using paired *t*-tests. (Keys: *: significant at *p* ≤ 0.05 (between-group comparison with placebo); ^#^: significant at *p* ≤ 0.05 (within-group comparison with baseline); ^##^: significant at *p* ≤ 0.01 (within-group comparison with baseline)).

**Table 4 nutrients-16-01211-t004:** Results of POMS-2 questionnaire.

		Baseline	Week 8	Week 12	Change from Baseline	
					Week 8	Week 12
Total Mood Disturbance	Placebo (*n* = 31)	42.8 ± 6.2	43.2 ± 7.0	43.4 ± 7.0	0.4 ± 3.6	0.6 ± 3.2
	PHGG (*n* = 30)	43.3 ± 6.7	42.6 ± 6.0	42.4 ± 7.2	−0.8 ± 3.4	−0.9 ± 4.1
	*p*-value	0.737	0.226	0.099	0.226	0.099
Anger-Hostility	Placebo (*n* = 31)	44.5 ± 6.3	44.6 ± 6.8	44.8 ± 6.8	0.1 ± 4.5	0.3 ± 4.2
	PHGG (*n* = 30)	43.2 ± 4.8	43.2 ± 4.8	42.8 ± 6.0	0.1 ± 3.4	−0.3 ± 4.6
	*p*-value	0.351	0.792	0.459	0.792	0.459
Confusion-Bewilderment	Placebo (*n* = 31)	45.6 ± 7.2	46.1 ± 7.8	45.5 ± 7.6	0.4 ± 5.0	−0.2 ± 4.1
	PHGG (*n* = 30)	47.4 ± 8.3	45.7 ± 7.0	45.2 ± 7.7 ^#^	−1.8 ± 5.1	−2.3 ± 5.1
	*p*-value	0.370	0.142	0.141	0.142	0.141
Depression-Dejection	Placebo (*n* = 31)	45.7 ± 5.4	45.3 ± 5.1	45.7 ± 5.2	−0.4 ± 3.1	0.0 ± 3.0
	PHGG (*n* = 30)	45.0 ± 4.4	44.8 ± 4.1	44.5 ± 4.5	−0.3 ± 3.7	−0.6 ± 3.1
	*p*-value	0.611	0.959	0.368	0.959	0.368
Fatigue-Inertia	Placebo (*n* = 31)	41.4 ± 5.0	42.5 ± 6.5	42.7 ± 6.4	1.1 ± 5.2	1.3 ± 5.0
	PHGG (*n* = 30)	43.2 ± 7.7	43.2 ± 6.2	42.4 ± 7.8	0.1 ± 4.9	−0.8 ± 4.6
	*p*-value	0.301	0.725	0.134	0.725	0.134
Tension-Anxiety	Placebo (*n* = 31)	44.2 ± 6.1	44.7 ± 6.9	44.9 ± 7.9	0.5 ± 4.4	0.7 ± 4.5
	PHGG (*n* = 30)	43.7 ± 7.3	44.1 ± 7.4	43.9 ± 7.9	0.4 ± 5.0	0.2 ± 5.3
	*p*-value	0.777	0.825	0.654	0.825	0.654
Vigor-Activity	Placebo (*n* = 31)	57.1 ± 9.9	56.4 ± 9.6	56.4 ± 9.0	−0.8 ± 6.4	−0.7 ± 5.2
	PHGG (*n* = 30)	54.7 ± 10.7	57.1 ± 9.8 ^#^	55.3 ± 10.4	2.4 ± 6.2	0.6 ± 6.4
	*p*-value	0.369	0.080	0.569	0.080	0.569
Friendship	Placebo (*n* = 31)	59.7 ± 8.6	59.8 ± 9.6	59.3 ± 8.6	0.1 ± 7.1	−0.5 ± 6.5
	PHGG (*n* = 30)	57.8 ± 9.5	58.1 ± 8.5	58.0 ± 10.0	0.3 ± 6.6	0.2 ± 8.6
	*p*-value	0.415	0.827	0.999	0.827	0.999

Data are represented as mean ± SD. Between-group comparisons of the baseline were conducted using the Student’s *t*-test, while comparisons at weeks 8 and 12 were analyzed using a linear model that included baseline values as covariates, along with time points, groups, interactions between time points and groups, interactions between baseline values and time points, and participants as factors. Within-group comparisons were performed using paired *t*-tests. (Keys: ^#^: significant at *p* ≤ 0.05 (within-group comparison with baseline)). No significant differences were detected between the placebo and PHGG groups.

## Data Availability

Data is contained within the article.

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
