# Peer review of "Effectiveness of Partially Hydrolyzed Guar Gum on Cognitive Function and Sleep Efficiency in Healthy Elderly Subjects in a Randomized, Double-Blind, Placebo-Controlled, and Parallel-Group Study"

_nutrients, 2024, doi:10.3390/nu16081211_

Round 1

Reviewer 1 Report

Comments and Suggestions for Authors

The aim of the manuscript entitled „Effectiveness of partially hydrolyzed guar gum on cognitive function, sleep efficiency in healthy elderly subjects in a randomized, double-blind, placebo-controlled, and parallel-group study for the prevention of frailty” was to examines the effects of water-soluble prebiotic dietary fiber, partially hydrolyzed guar gum (PHGG), on cognitive function and mental health in healthy elderly individuals. The aging of the population across the world is connected with the increasing interest in the prevention of age-related disabilities, including cognitive disorders. The background of the study was properly described, but in my opinion, the authors use many logical shortcuts, starting the introduction with frailty syndrome and moving on to dementia, even though this is not the subject of research. I would focus more on the description of cognitive impairment and mild cognitive impairment because the study concerns healthy older people who may also experience the effects of changes in the brain. It should be noted that also the title of the manuscript refers to a frailty syndrome that has not been studied.

The research methodology was well-planned and described. My only question is whether MMSE was in the Japanese version - it is not clear from the description (MMSE-J). It would be beneficial for the reader to describe Cognitrax tests in more detail, as it is not a widely used tool for assessing cognitive functions.

My biggest reservations concern the statistical analyses performed because the tables with the results were not described in detail. Both in the column comparing points at study time points and in the column comparing changes from baseline we see exactly the same p-values, which makes me doubt the correctness of the statistical method used; if the authors had described the statistical methods used in the methodology or the table description, everything would have been obvious. It would also be easier for the reader if the table description included information that the compared data were standardized.

In Table 3, I also question the small difference in values for which statistically significant differences were demonstrated (week 8), while in baseline the difference in points was larger and the difference was not significant. Can authors explain this?

In the case of Table 4, it is an abuse to indicate improvement when the result is not significant - it would be beneficial to clearly emphasize that there are no differences.

The conclusions of the work are consistent with the obtained results.

Author Response

Reviewer #1:

  • The background of the study was properly described, but in my opinion, the authors use many logical shortcuts, starting the introduction with frailty syndrome and moving on to dementia, even though this is not the subject of research. I would focus more on the description of cognitive impairment and mild cognitive impairment because the study concerns healthy older people who may also experience the effects of changes in the brain. It should be noted that also the title of the manuscript refers to a frailty syndrome that has not been studied.

Response:

We are thankful to this reviewer for the valuable and important remarks. According to the reviewer's advice, we have omitted the word "frailty" from the manuscript title and revised the background information in the introduction section of the revised version of the manuscript (please see pages 1-2).

  • My only question is whether MMSE was in the Japanese version – it is not clear from the description (MMSE-J). It would be beneficial for the reader to describe Cognitrax tests in more detail, as it is not a widely used tool for assessing cognitive functions.

Response:

We used the Japanese version of MMSE and corrected the description as (MMSE-J) in the manuscript text under the heading Study Participants (please see Section 2.3; page 3) and included reference No. 37. We agreed with the reviewer’s comment on Cognitrax; therefore, we have added a detailed description of the Cognitrax test in Section 2.5 (see pages 3 to 4).

  • My biggest reservations concern the statistical analyses performed because the tables with the results were not described in detail. Both in the column comparing points at study time points and in the column comparing changes from baseline we see exactly the same p-values, which makes me doubt the correctness of the statistical method used; if the authors had described the statistical methods used in the methodology or the table description, everything would have been obvious. It would also be easier for the reader if the table description included information that the compared data were standardized.

         In Table 3, I also question the small difference in values for which statistically           significant differences were demonstrated (week 8), while in baseline the                 difference in points was larger and the difference was not significant. Can               authors explain this?

Response:

We highly appreciate the reviewer's concerns. In the tables representing the Cognitrax results, OSA-MA sleep questionnaire, and POMS-2 questionnaire results, the baseline comparison between groups was performed using a Student's t-test to check the homogeneity between the assigned groups. Whereas, the comparisons at week 8 and after week 12 of intakes between groups were conducted using a mixed linear model using baseline values as covariates for time points, groups, and between time points and group interactions as factors.

The results of the group comparisons for both actual values and estimated changes from baseline are not very different since both analyses used baseline scores as covariates. We would like to emphasize that the between-group comparisons are not just simple comparisons of mean values but are adjusted factors with baseline values. Therefore, even if the actual mean differences between groups are modest, significant differences can still prevail, unlike the baseline score comparisons using the Student's t-test.

We revised the manuscript in Section 2.9 of the statistical analysis (page 5). We have added detailed descriptions of the statistical analysis methods at the bottom of each table for easier understanding by the readers.

  • In the case of Table 4, it is an abuse to indicate improvement when the result is not significant - it would be beneficial to clearly emphasize that there are no differences.

Response:

We apologize for not mentioning the conclusive details in this table. In the revised version of the manuscript, we added the statement 'No significant differences were detected between the placebo and PHGG groups' in the note under Table 4 (please see page 10). In addition, we have revised the description of the results in Section 3.4 of the POMS-2 Questionnaire (page 9), as well as the discussion regarding the POMS-2 results in the revised text of the manuscript (page 11).

Reviewer 2 Report

Comments and Suggestions for Authors

The manuscript is an interesting piece of work. However some doubts are to be mentioned, and some sentences – clarified/reedited. Please see them below:”

1)      „The population and proportion of the elderly are also increasing globally. and (…)” - this part of the sentence needs to be corrected

2)      „The participants took either PHGG or a placebo every day with water for 12 consecutive weeks”

Was the time of day appropriate to consume preparation indicated?

Was the time of the preparation intake - after/before a meal – indicated?

Did the participants provide this information?

It should be also widely discussed in the appropriate part of the manuscript – please add.

3)      The explanation of sample size isn’t clear for me (see 2.7. Sample size)

The Authors determined the sample size „by referencing studies related to improvements in gut health and mental well-being”, and they required around 40-80 participants/group.

Meanwhile they settled on a sample size of 30 (33)/group.

Could you explain the reason? This text is not clear for now.

4)      The research results are interesting, but the discussion needs to be organized and some important aspects - added.

One issue to discuss is the experimental period (why did the Authors choose 12 weeks), and what was the reason for the examination also after 8 weeks.

The part about research involving children is unnecessary, but the comparison of the obtained results to other research involving elderly is, in my opinion, crucial.

 It is difficult to base the results on the effect of the preparation on the intestinal microflora when stool examination was not performed. Some important neurotransmiter levels weren’t examined, too. Thus, I do not understand how the Authors wanted to explain the results obtained? In my opinion, without any parameters examined to explain the potential mechanisms,  another study, with the same conditions in needed.

5)      The mechanisms that could have caused such results, even in a short-term study, should be discussed one by one, also providing references to the literaturÄ™ sources. If the discussion also includes studies on animal models, it is necessary to indicate the age of the animals, the dose and the duration of the experiment. Of course, appropriate information should also be provided and discussed regarding research involving humans.

6)      Conclusions should be based, in my opinion, on what exactly results (statistically significant differences) showed.

Author Response

Reviewer #2:

  • The population and proportion of the elderly are also increasing globally. and (…)” - this part of the sentence needs to be corrected

Response:

We are grateful to the reviewer for pointing out the typos, which were corrected. Further, we have carefully checked the manuscript text thoroughly and corrected the phases in the revised manuscript.

  • The participants took either PHGG or a placebo every day with water for 12 consecutive weeks”

Was the time of day appropriate to consume preparation indicated?

Was the time of the preparation intake - after/before a meal – indicated?

Did the participants provide this information?

It should be also widely discussed in the appropriate part of the manuscript – please add.

Response:

We appreciate the reviewer’s constructive feedback on this section of the manuscript. We added the required description of the timing of the intakes in the revised manuscript (please see Section 2.4, page 3).

  • The explanation of sample size isn’t clear for me (see 2.7.Sample size)

The Authors determined the sample size by referencing studies related to improvements in gut health and mental well-being, and they required around 40-80 participants/group.

Meanwhile they settled on a sample size of 30 (33)/group.

Could you explain the reason? This text is not clear for now.

Response:

We apologize for the confusion. After careful reading of the manuscript, we have corrected the phrase in Section 2.7 of Sample Size (page 4).

  • One issue to discuss is the experimental period (why did the Authors choose 12 weeks), and what was the reason for the examination also after 8 weeks.

The part about research involving children is unnecessary, but the comparison of the obtained results to other research involving elderly is, in my opinion, crucial.

It is difficult to base the results on the effect of the preparation on the intestinal microflora when stool examination was not performed. Some important neurotransmiter levels weren’t examined, too. Thus, I do not understand how the Authors wanted to explain the results obtained? In my opinion, without any parameters examined to explain the potential mechanisms, another study, with the same conditions in needed.

Response:

Thank you again for the constructive comment to improve the quality of our manuscript. Concerning the experimental duration, we have included a phrase describing the reason in the revised manuscript (please see Section 2.4, page 3).

We agree with the reviewer's statement that the present clinical trial did not measure the impact on the intestinal environment or brain neurotransmitters. However, it was necessary to point out the hypothesized mechanisms behind the effects observed in this clinical trial; therefore, we stated the information inferred from the available literature on clinical trials on the elderly and ex-vivo tests using feces from elderly subjects. As suggested by the reviewer, we have omitted the irrelevant information from studies on children in the revised version of the manuscript (please see the highlighted areas on pages 11 and 12).

  • The mechanisms that could have caused such results, even in a short-term study, should be discussed one by one, also providing references to the literature sources. If the discussion also includes studies on animal models, it is necessary to indicate the age of the animals, the dose and the duration of the experiment. Of course, appropriate information should also be provided and discussed regarding research involving humans.

Response:

As per the reviewer's advice, we have indicated the details about participant ages in human trials, animal ages in animal trials, and intake durations in the revised manuscript (please see the highlighted areas on pages 11 and 12).

  • Conclusions should be based, in my opinion, on what exactly results (statistically significant differences) showed.

Response:

Finally, according to the reviewer’s suggestion, we have removed the description of vitality from the Conclusion section (please see page 13) and provided a reasonable conclusion as advised.